# Impact of contralateral sensorineural hearing loss on prognosis in idiopathic sudden sensorineural hearing loss

Sanlin Xie[1,2☯], Zhifeng Chen[1,3☯], Lipeng Huang[4], Yongjun Hong[2]*, Chang Lin[1,3]*

1 Department of Otorhinolaryngology Head and Neck Surgery, The First Affiliated Hospital of Fujian Medical University, Fuzhou, China, 2 Department of Otolaryngology Head and Neck Surgery, Zhongshan Hospital of Xiamen University, School of Medicine, Xiamen University, Xiamen, China, 3 Department of Otorhinolaryngology Head and Neck Surgery, National Regional Medical Center, Binhai Campus of the First Affiliated Hospital, Fujian Medical University, Fuzhou, China, 4 Department of thoracic, Xiang'an Hospital of Xiamen University, School of Medicine, Xiamen University, Xiamen, China

☯ These authors contributed equally to this work.
* 13959223003@163.com (YH), linc301@sina.com (CL)

**Data availability statement:** All data are available from the Figshare database (URLs:https://doi.org/10.5281/zenodo.18268353).

**Funding:** This research received external funding by Fujian Provincial Science and Technology Innovation Platform (grant number:2021Y2002).

## Abstract

### Objective

The aim of this study was to investigate the impact of contralateral sensorineural hearing loss on prognosis in idiopathic sudden sensorineural hearing loss.

### Methods

A retrospective analysis was conducted on 445 ISSNHL patients treated between January 2020 and January 2024. The patients were divided into two groups: recovery (234 cases, including complete and partial recovery) and no-recovery (211 cases, no recovery). Pure-tone audiometry was used for evaluation. Clinical characteristics were compared between the groups, and multivariate logistic regression was performed to identify risk factors for poor prognosis.

### Results

Significant differences were observed between the two groups regarding contralateral hearing level, age, duration of illness, diabetes, hypertension, and audiometric curves. An increase in the hearing threshold of the contralateral ear was positively correlated with the risk of ineffective treatment. Patients with contralateral sensorineural hearing loss prior to the onset of ISSNHL exhibited a 2.757-fold increased risk of treatment failure compared with ISSNHL patients who had normal contralateral ear hearing. Multivariate logistic regression analysis revealed that contralateral sensorineural hearing loss exhibited a significant multiplicative interaction with age over 60 years, hypertension, and a disease duration exceeding 4 days.

**Competing interests:** The authors have declared that no competing interests exist.

## Conclusion

The presence of contralateral sensorineural hearing loss before the onset of ISSNHL is an independent risk factor for poor prognosis in ISSNHL. It interacts significantly with age over 60 years, hypertension, and disease duration exceeding 4 days. A thorough assessment of contralateral hearing status, in conjunction with factors such as age, hypertension, and disease duration, is essential for developing personalized treatment plans to improve prognosis.

## Introduction

Idiopathic sudden sensorineural hearing loss (ISSNHL) is a sudden hearing loss of unknown cause, characterized by a rapid decline in hearing to its lowest level within 72 hours. Pure-tone average (PTA) reveals thresholds exceed 30 dB across three consecutive frequencies [1]. Epidemiological data indicate the annual incidence of ISSNHL ranges from 5–27 per 100,000 individuals in the U.S. population, corresponding to approximately 4,000–25,000 new diagnoses each year [2]. Clinical outcomes vary widely, with some patients experiencing complete or partial recovery, while others develop persistent auditory dysfunction that significantly compromises daily functioning. For patients whose contralateral already exhibits hearing loss prior to the onset of ISSNHL, even minimal improvement in the affected ear may hold considerable clinical value.

However, current clinical practice and research predominantly focus on the ear affected at presentation, often overlooking the hearing status of the contralateral ear and its potential influence on prognosis. In fact, presence of contralateral sensorineural hearing loss(SNHL) before the onset of ISSNHL may have a meaningful impact on treatment outcomes. Despite this, limited research has explored whether such contralateral SNHL influences the final prognosis [3,4].

Building upon existing evidence [3,4], the present study retrospectively analyzed 445 cases of ISSNHL to investigate associations between various prognostic factors and treatment outcomes, with particular emphasis on the role of presence contralateral SNHL. These findings aim to inform clinical decision-making by providing empirical support for comprehensive, individualized patient assessment protocols.

## Methods

### Patients

This study employed a retrospective case-control design, involving 445 patients diagnosed with ISSNHL who were hospitalized and treated between 2020 and 2024. Comprehensive clinical data were collected, including age, gender, affected ear, presence of diabetes or hypertension, accompanying symptoms such as tinnitus, vertigo, and aural fullness, smoking history, duration of illness (defined as the interval from symptom onset to initiation of treatment), hearing thresholds, audiometric curves, and degree of hearing loss.

Inclusion criteria:sudden onset of sensorineural hearing loss within 72 hours, hearing loss >30 dB affecting at least three contiguous frequencies on pure-tone audiometry;first-time unilateral occurrence;symptom duration ≤14 days and no prior treatment(patients referred to those receiving their first treatment at our hospital after symptom onset, with no prior interventions administered elsewhere);age ≥ 18 years.

Exclusion criteria:presence of middle ear pathology, inner ear malformations, or vestibular schwannoma (in either ear); used ototoxic medications within the past month; age < 18 years;pregnancy or lactation;history of ear surgery;contralateral conductive or mixed hearing loss;incomplete clinical data.

This study is a retrospective analysis approved by the Ethics Committee of Zhongshan Hospital of Xiamen University (approval number: 2024−164). All data were fully de-identified, and informed consent from participants was not required. The Ethics Committee waived the requirement for informed consent.

The data for this study were accessed for research purposes from 25/12/2024–15/01/2025. During the data access period, the authors did not have access to information that could identify individual participants.

## Hearing assessment

Patients underwent medical history, lab tests, audiometric evaluation, and MRI. PTA was calculated as the average of the air-conduction thresholds at 0.5, 1, 2, and 4 kHz for the affected ear. The severity of hearing loss was classified as follows: mild (26 ≤ PTA < 41 dB), moderate (41 ≤ PTA < 61 dB), severe (61 ≤ PTA < 81 dB), and profound (PTA ≥ 81 dB). audiometric curves were categorized into four types based on the hearing threshold patterns [5]:

Low-frequency type: Hearing loss primarily occurs at frequencies ≤ 1 kHz, with thresholds at 0.25 kHz and 0.5 kHz showing a hearing loss of ≥ 20 dB.

High-frequency type: Hearing loss primarily occurs at frequencies ≥ 2 kHz, with thresholds at 4 kHz and 8 kHz showing a hearing loss of ≥ 20 dB.

Flat type: Relatively uniform hearing loss is observed across all frequencies from 0.25 to 8 kHz (at 0.25, 0.5, 1, 2, 4, and 8 kHz), with a mean PTA ≤ 80 dB.

Total deafness type: Profound hearing loss is observed across all tested frequencies (0.25–8 kHz), with a mean PTA ≥ 81 dB.

The degree of hearing improvement was assessed based on the change in PTA of the affected ear following treatment. The criteria were defined as follows:complete recovery: hearing in the affected ear returns to normal levels or is within 10 dB of the affected ear; partial recovery: hearing in the affected ear improves to within 50% of pre-onset levels or shows a gain of more than 10 dB; no recovery: hearing in the affected ear improves by less than 10 dB [1].Based on these definitions, patients were categorized into two groups:recovery group: includes patients with complete or partial recovery. no-recovery: includes patients with no recovery.

## Contralateral hearing assessment

The hearing status of the contralateral ear was evaluated according to the following criteria:

1. contralateral SNHL: defined as contralateral ear PTA > 25 dB;

2. contralateralear normal hearing: PTA ≤ 25 dB;

3. conductive or mixed hearing loss were excluded;

4. exclusion of bilateral ISSNHL: simultaneous hearing loss in both ears, defined as PTA > 30 dB at three consecutive frequencies in the contralateral ear;or cases in which the contralateral ear exhibited a hearing threshold shift >10 dB before and after treatment.

## Treatment methods

All patients received systemic corticosteroid therapy upon hospital admission, primarily using dexamethasone sodium phosphate injections. The treatment protocol was as follows: an intravenous dose of 10 mg was administered once daily for the first three consecutive days, followed by a reduced dose of 5 mg on the fourth and fifth days. Beginning on the second day of treatment, patients also received intratympanic injections of dexamethasone sodium phosphate. Each intratympanic dose was 0.5 mL, administered every other day for a total of four sessions. The entire treatment course lasted 10 days. For patients who achieved complete hearing recovery by the time of discharge, PTA recorded at discharge was used for evaluation. In patients without complete recovery, PTA values obtained during the 1-month follow-up visit were used instead.

## Statistical methods

The chi-square test, Fisher's exact test, or Wilcoxon rank-sum test was used to compare demographic characteristics and relevant clinical variables between groups. The Wilcoxon rank-sum test was specifically employed to assess differences in contralateral hearing levels between the recovery and no-recovery groups, and violin plots were generated for visualization.

An unconditional logistic regression model was used to calculate odds ratios (ORs) and 95% confidence intervals (CIs) to evaluate the association between contralateral hearing levels and treatment efficacy. Multivariate logistic regression analyses were performed to identify independent prognostic factors among those found to be significant in univariate analysis. To assess model performance, the Akaike Information Criterion (AIC) was calculated to compare the goodness-of-fit of different multivariate models. Receiver Operating Characteristic (ROC) curves were also constructed to evaluate the predictive accuracy of the models.

All statistical analyses were conducted using R software (version 4.2.2), and a two-sided p-value < 0.05 was considered statistically significant.

## Results

Among the 445 patients with ISSNHL, 234 were classified into the recovery group and 211 into the no-recovery group. The variables, including age, duration of illness, pre-treatment PTA of the affected ear, and PTA of the contralateral ear, did not follow a normal distribution; therefore, non-parametric tests were employed. No significant differences were observed between the two groups in terms of gender, side of the affected ear, presence of tinnitus, vertigo, aural fullness, smoking history, pre-treatment PTA of the affected ear, or degree of hearing loss (all p > 0.05). However, significant differences were found in age (p < 0.001) and duration of illness (p = 0.011). Comorbidities were more common in the no-recovery group, including diabetes (27.49% vs. 16.24%, p = 0.004) and hypertension (33.65% vs. 21.79%, p = 0.005). Audiometric curve types also differed significantly between groups (p = 0.007). Detailed comparisons are presented in Table 1.

As illustrated in Fig 1, the median PTA value of the contralateral ear was 18 dB in the recovery group and 26 dB in the no-recovery group, representing a statistically significant difference (p < 0.001). When analyzed as a continuous variable, contralateral hearing level was positively correlated with the risk of treatment failure in ISSNHL patients (OR: 1.024; 95% CI: 1.014–1.034). Furthermore, when contralateral hearing was categorized into normal and SNHL, and potential confounders were controlled for, contralateral SNHL remained significantly associated with an increased risk of treatment failure. Patients with contralateral SNHL had a 2.757-fold higher risk of treatment failure compared to those with normal hearing (95% CI: 1.689–4.499) (Table 2).

Furthermore, data-driven analyses revealed that contralateral SNHL exhibits significant multiplicative interactions with age > 60 years, hypertension, and a disease duration >4 days, all of which were associated with an increased risk of treatment failure(Table 3).

**Table 1. Comparison of characteristics between the recovery and no-recovery group.**

| Variables | Outcome of treatment | | P |
|---|---|---|---|
| | Recovery (N = 234) | No-recovery (N = 211) | |
| Gender (Male/Female n (%)) | | | 0.756 |
| Male | 113(51.83) | 105(48.17) | |
| Female | 121(53.30) | 106(46.70) | |
| Age (years, M ($P_{25}$, $P_{75}$)) | 46.5(35,56) | 52(42,62) | <0.001* |
| Smoking (n (%)) | | | 0.937 |
| No | 220(52.63) | 198(47.37) | |
| Yes | 14(51.85) | 13(48.15) | |
| Diabetes (n (%)) | | | 0.004* |
| No | 196(56.16) | 153(43.84) | |
| Yes | 38(39.58) | 58(60.42) | |
| Hypertension (n (%)) | | | 0.005* |
| No | 183(56.66) | 140(43.34) | |
| Yes | 51(41.80) | 71(58.20) | |
| duration of illness (d, M ($P_{25}$, $P_{75}$)) | 4(2,7) | 5(3,8) | 0.011* |
| affected side (n (%)) | | | 0.374 |
| Right | 111(50.45) | 109(49.55) | |
| Left | 123(54.67) | 102(45.33) | |
| Tinnitus (n (%)) | | | 0.816 |
| No | 25(51.02) | 24(48.98) | |
| Yes | 209(52.78) | 187(47.22) | |
| Vertigo (n (%)) | | | 0.881 |
| No | 147(52.31) | 134(47.69) | |
| Yes | 87(53.05) | 77(46.95) | |
| Aural fullness (n (%)) | | | 0.481 |
| No | 188(51.79) | 175(48.21) | |
| Yes | 46(56.10) | 36(43.90) | |
| audiometric curves (n (%)) | | | 0.007* |
| Low frequency | 31(77.50) | 9(22.50) | |
| High frequency | 8(40.00) | 12(60.00) | |
| Flat-type frequency | 90(49.45) | 92(50.55) | |
| Total deafness | 105(51.72) | 98(48.28) | |
| Level of hearing loss (n (%)) | | | 0.445 |
| Mild | 28(51.85) | 26(48.15) | |
| Moderate | 40(46.51) | 46(53.49) | |
| Severe | 64(58.18) | 46(41.82) | |
| Profound | 102(52.31) | 93(47.69) | |
| Pre-treatment PTA (affected ear) (dB M ($P_{25}$, $P_{75}$)) | 77(55,100) | 78(57,107) | 0.403 |
| Pre-treatment PTA (contralateral) (dB M ($P_{25}$, $P_{75}$)) | 18(14,25) | 26(17,43) | <0.001 * |
| Post-treatment PTA (affected ear) (dB M ($P_{25}$, $P_{75}$)) | 35(21.75,59.25) | 77(55.100) | <0.001 * |
| Post-treatment PTA (contralateral) (dB M ($P_{25}$, $P_{75}$)) | 18(13,25) | 25(17,42) | <0.001 * |

PTA: pure tone average; *Differences were considered statistically significant if $p < 0.05$ (*$p < 0.05$)

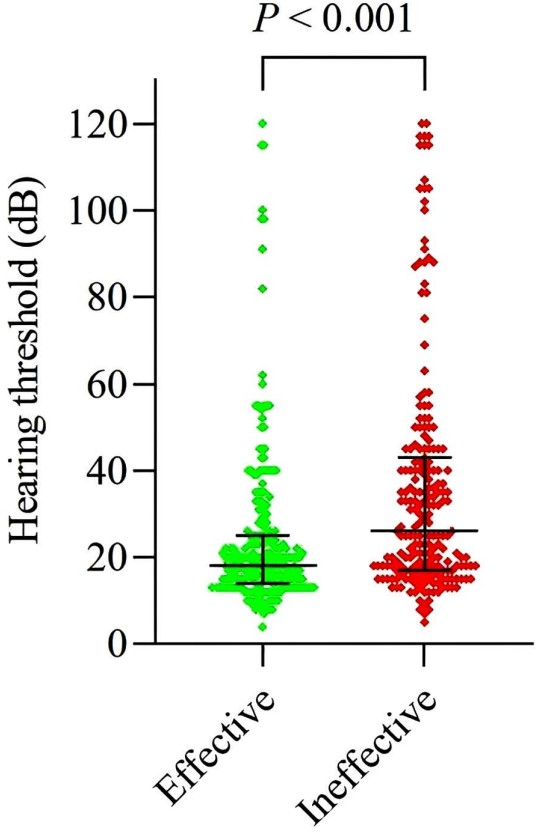

**Fig 1. Distribution of the contralateral hearing threshold in Recovery and No-recovery groups.** Green dots represent the contralateral hearing thresholds of the Recovery group (n = 234), and red dots represent the contralateral hearing thresholds of the No-recovery group (n = 211). Each dot corresponds to an individual subject. The difference between the two groups was compared using the Mann-Whitney U test, and the result was statistically significant (P < 0.001).

**Table 2. *OR* for the hearing threshold of the contralateral in relation to treatment failure.**

| Variables | Outcome of treatment | | OR (95%CI) | OR (95%CI)[a] |
|---|---|---|---|---|
| | Recovery (%) | No-recovery (%) | | |
| Hearing threshold (continuous) | 234(100) | 211(100) | 1.024(1.014,1.034) | 1.021(1.010,1.031) |
| Hearing threshold (contralateral) | | | | |
| Normal | 177(75.64) | 105(49.76) | 1.000 | 1.000 |
| SNHL | 57(24.36) | 106(50.24) | 3.135(2.096,4.688) | 2.757(1.689,4.499) |

[a]Adjusted for age, gender, smoking, diabetes, hypertension, duration of illness, affected side, tinnitus, vertigo, aural fullness, audiometric curves, level of hearing loss and hearing threshold of affected ear

Multivariate logistic regression analysis was performed to further evaluate prognostic factors. Model 1 included age, diabetes, hypertension, duration of illness, and audiometric curve type. Model 2 was constructed by adding contralateral hearing level to the variables in Model 1 (Table 4). Compared to Model 1, Model 2 showed improved model fit, indicated by a lower Akaike Information Criterion (AIC) value (588.467), and better discriminative ability, as reflected by a larger area under the receiver operating characteristic (ROC) curve (Fig 2).

**Table 3. Analysis of the influence of contralateral hearing threshold with age, diabetes, hypertension, duration of illness and audiometric curves on outcome of treatment.**

| Variables | | Outcome of treatment | | OR (95% CI)[a] | P |
|---|---|---|---|---|---|
| | | Recovery (%) | No-recovery(%) | | |
| Hearing threshold (contralateral) | Age (years) | | | | |
| Normal | < 60 | 156(63.41) | 90(36.59) | 1.000 | |
| Normal | ≥ 60 | 21(58.33) | 15(41.67) | 1.010(0.464,2.203) | 0.979 |
| SNHL | < 60 | 32(36.36) | 56(63.64) | 2.916(1.672,5.085) | <0.001 |
| SNHL | ≥ 60 | 25(33.33) | 50(66.67) | 3.065(1.643,5.719) | <0.001 |
| Hearing threshold (contralateral) | Diabetes | | | | |
| Normal | No | 159(65.16) | 85(34.84) | 1.000 | |
| Normal | Yes | 18(47.37) | 20(52.63) | 1.626(0.773,3.418) | 0.200 |
| SNHL | No | 37(35.24) | 68(64.76) | 3.231(1.864,5.601) | <0.001 |
| SNHL | Yes | 20(34.48) | 38(65.52) | 2.692(1.295,5.599) | 0.008 |
| Hearing threshold (contralateral) | Hypertension | | | | |
| Normal | No | 145(65.91) | 75(34.09) | 1.000 | |
| Normal | Yes | 32(51.61) | 30(48.39) | 1.578(0.815,3.055) | 0.176 |
| SNHL | No | 38(36.89) | 65(63.11) | 2.917(1.637,5.195) | <0.001 |
| SNHL | Yes | 19(31.67) | 41(68.33) | 3.866(1.786,8.367) | 0.001 |
| Hearing threshold (contralateral) | duration of illness | | | | |
| Normal | < 4 | 83(66.40) | 42(33.60) | 1.000 | |
| Normal | ≥ 4 | 94(59.87) | 63(40.13) | 1.292(0.763,2.186) | 0.341 |
| SNHL | < 4 | 25(41.67) | 35(58.33) | 2.279(1.113,4.664) | 0.024 |
| SNHL | ≥ 4 | 32(31.07) | 71(68.93) | 4.063(2.156,7.658) | <0.001 |
| Hearing threshold (contralateral) | audiometric curves | | | | |
| Normal | Non-total deafness | 99(67.35) | 48(32.65) | 1.000 | |
| Normal | Total deafness | 78(57.78) | 57(42.22) | 1.386(0.413,4.652) | 0.597 |
| SNHL | Non-total deafness | 30(31.58) | 65(68.42) | 4.305(2.293,8.085) | <0.001 |
| SNHL | Total deafness | 27(39.71) | 41(60.29) | 1.996(0.539,7.398) | 0.301 |

[a]Adjusted for age, gender, smoking, diabetes, hypertension, duration of illness, affected side, tinnitus, vertigo, aural fullness, audiometric curves, level of hearing loss and hearing threshold of affected ear.

The patients were divided into two groups: contralateral ear normal hearing group and contralateral ear SNHL group. The median hearing threshold of the contralateral normal hearing group was 17 dB, while that of the contralateral SNHL group was 40 dB. A statistically significant difference was found between the two groups in contralateral hearing thresholds (p<0.001). No significant difference in the initial hearing thresholds of the affected ear was observed between the two groups. After treatment, the treatment outcome in the contralateral normal hearing group was significantly better than that in the contralateral SNHL group (p<0.001), while the hearing gain in the contralateral SNHL group was significantly lower than that in the contralateral normal hearing group (p<0.001).(Table 5).

## Discussion

ISSNHL is a condition that exhibits susceptibility across all age groups, with its incidence rate increasing with age [6]. The onset of symptoms is rapid, and the disease progresses quickly, often presenting with hearing loss, tinnitus, and vertigo. The underlying pathological mechanisms remain incompletely understood, and in addition to the auditory symptoms, patients frequently experience significant psychological distress, including anxiety and fear. These factors

 

**Table 4. Multivariate unconditional logistic regression analysis of influencing factors of outcome of treatment.**

| Variables | Model1[a] | | Model2[b] | |
|---|---|---|---|---|
| | OR (95% CI) | P | OR (95% CI) | P |
| Age (years) | | | | |
| < 60 | 1.000 | | 1.000 | |
| ≥ 60 | 1.449(0.902,2.326) | 0.125 | 1.000(0.599,1.670) | 1.000 |
| Diabetes | | | | |
| No | 1.000 | | 1.000 | |
| Yes | 1.637(1.003,2.670) | 0.048 | 1.316(0.790,2.192) | 0.292 |
| Hypertension | | | | |
| No | 1.000 | | 1.000 | |
| Yes | 1.500(0.945,2.382) | 0.085 | 1.505(0.936,2.420) | 0.092 |
| duration of illness | | | | |
| < 4 | 1.000 | | 1.000 | |
| ≥ 4 | 1.559(1.055,2.303) | 0.026 | 1.473(0.988,2.198) | 0.057 |
| audiometric curves | | | | |
| Non-total deafness | 1.000 | | 1.000 | |
| Total deafness | 0.973(0.658,1.437) | 0.889 | 1.088(0.727,1.627) | 0.682 |
| Hearing threshold (contralateral) | | | | |
| Normal | | | 1.000 | |
| SNHL | | | 2.776(1.780,4.331) | <0.001 |
| AIC | 607.306 | | 588.467 | |

[a]Adjusted for age, diabetes, hypertension, duration of illness and audiometric curves

[b]Additional adjustment of hearing threshold of contralateral.

can severely impact the patient's quality of life and work efficiency, thereby further exacerbating the clinical burden of ISSNHL [7].

Studies have shown that the treatment outcomes of ISSNHL are influenced by various factors, including age, initial hearing level, duration of illness, degree of hearing loss, as well as comorbidities such as hypertension and diabetes [8–11]. However, research investigating the impact of contralateral SNHL on the prognosis of ISSNHL remains limited.

The study found no significant difference in the pre-treatment hearing thresholds of the affected between the recovery and no-recovery groups. However, the contralateral in the recovery group exhibited significantly lower hearing thresholds compared to those in the no-recovery group. Notably, the proportion of patients with hearing loss in the contralateral ear was significantly higher in the no-recovery group (50.24% vs 24.36%) than in the recovery group, with a statistically significant difference (p < 0.05).

Multivariate logistic regression analysis further confirmed that an increase in the contralateral SNHL was significantly positively correlated with an increased risk of treatment failure in ISSNHL. After adjusting for potential confounders, the results indicated that ISSNHL patients with contralateral SNHL before the onset of ISSNHL had a 2.757-fold higher risk of treatment failure compared to those with normal contralateral hearing (95% CI: 1.689, 4.499). Moreover, the study developed a prognostic prediction model for ISSNHL using multivariate logistic regression analysis. In constructing the model, we incorporated key variables, including age, diabetes, hypertension, audiometric curves, and disease duration, along with the significant variable of contralateral hearing level. The results revealed that the model incorporating contralateral hearing exhibited a lower AIC and a significantly larger area under the ROC curve, indicating superior discriminatory ability in predicting treatment efficacy. The findings suggest that presence of contralateral SNHL prior to the onset of ISSNHL may have poorer treatment outcomes and a significantly increased risk of treatment failure.

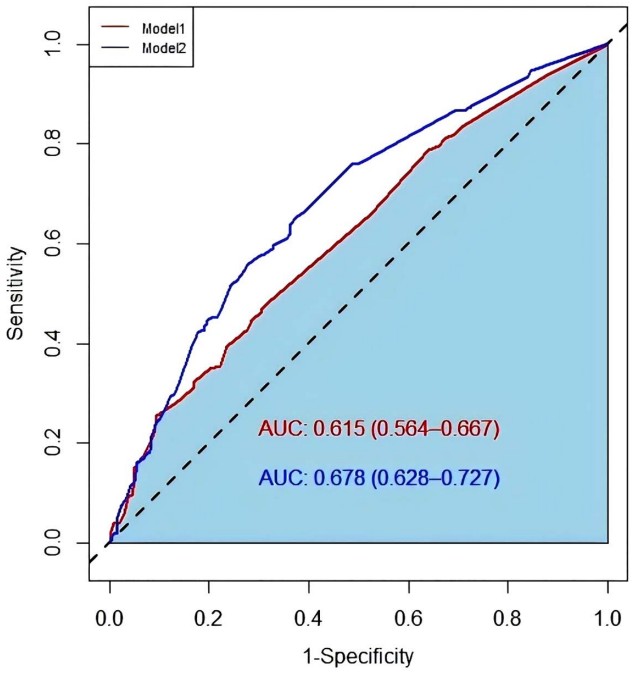

**Fig 2. Receiver Operating Characteristic curve of different multi factor models for predicting ISSNHL prognosis.** Fig 2 demonstrates the application of multivariate logistic regression analysis. Model 1 includes five variables: age, diabetes, hypertension, disease duration, and hearing curve. Model 2 adds the variable of contralateral hearing. The results indicate that the area under the ROC curve (AUC) for Model 2 is significantly larger (AUC: 0.678, 95% CI: 0.628-0.727).

**Table 5. Treatment outcomes of contralateral normal hearing group and contralateral SNHL group.**

| Variables | ISSNHL | | P |
|---|---|---|---|
| | contralateral normal hearing group(N = 282) | contralateral SNHL group(N = 163) | |
| Pre-treatment PTA (affected ear) (dB M ($P_{25}$, $P_{75}$)) | 78(53,101.25) | 75(60,103) | 0.265 |
| Pre-treatment PTA (contralateral) (dB M ($P_{25}$, $P_{75}$)) | 17(13,20) | 40(34,55) | <0.001* |
| Post-treatment PTA (affected ear) (dB M ($P_{25}$, $P_{75}$)) | 43(25,83) | 66(45,90) | <0.001* |
| Hearing gain(dB, M(P25, P75)) | 19(5,38) | 6(0,23) | <0.001* |
| Hearing recovery(n (%)) | | | <0.001* |
| Complete recovery | 86(30.5) | 22(16.6) | <0.001* |
| partial recovery | 91(32.3) | 35(18.4) | 0.015 |
| No recovery | 105(37.2) | 106(65.0) | <0.001* |

PTA: pure tone average; *Differences were considered statistically significant if $p < 0.05$ (*$p < 0.05$)

From a pathophysiological standpoint, cochlear hair cells are highly sensitive to hypoxia and ischemia. The blood supply to the cochlea is primarily provided by the terminal arteries, which lack collateral circulation. In combination with the high metabolic demands of cochlear hair cells, this anatomical feature renders them particularly vulnerable to damage

caused by hypoxia or ischemia. Consequently, contralateral SNHL may serve as an indicator of the cochlea's overall susceptibility to injury. This heightened susceptibility could be attributed to structural or metabolic characteristics of the cochlear tissues, which may, in turn, influence the cochlea's response to treatment.

Our study identified significant differences between the recovery and no-recovery groups in terms of age, diabetes, hypertension, duration of illness, and audiometric curves. Contralateral SNHL may independently or interactively influence the treatment outcomes of ISSNHL in conjunction with these factors. Subgroup analysis, after adjusting for other variables, revealed that contralateral SNHL exhibited significant multiplicative interactions with age over 60 years, hypertension, and a disease duration exceeding 4 days, all of which were associated with poorer prognosis in ISSNHL.

As demonstrated in previous studies and supported by our data, patient age serves as a reliable prognostic factor in ISSNHL cases [12–14]. Our findings further suggest that as patients age, their prognosis tends to worsen [15]. Additionally, ISSNHL patients with hypertension often present with more severe hearing loss and worse treatment outcomes [16,17]. We hypothesize that hypertension may indirectly compromise the blood supply to the inner ear by affecting vascular health and hemorheological properties, thus exacerbating contralateral hearing loss and increasing the likelihood of treatment inefficacy in ISSNHL. Regarding the impact of disease duration on the prognosis of ISSNHL, existing literature generally recognizes it as an important influencing factor [18–22]. A plausible explanation is that prolonged insufficient blood supply to the inner ear may lead to the progression of hair cell damage from functional to irreversible organic injury, thereby resulting in a worse prognosis. In conclusion, the interactions between contralateral sensorineural hearing loss and factors such as age, hypertension, and disease duration may influence the treatment outcomes of ISSNHL through various mechanisms.Previous studies have pointed out that the initial hearing threshold and degree of hearing loss are prognostic factors for SSNHL, with more severe hearing loss associated with poorer prognosis [10,23]; however, some studies have reached the opposite conclusion [24]. In our study, most of the included patients had moderate to profound hearing loss, which may have led to the lack of statistically significant differences between the two groups in terms of initial hearing levels and severity of hearing loss.

Additionally, this study found a positive correlation between an increase in the contralateral hearing threshold and the risk of treatment inefficacy. This suggests that patients with the one ear exhibiting SNHL may have a poorer prognosis if they experience sudden hearing loss in the other ear. Multiple studies have shown that ISSNHL patients often exhibit a higher prevalence of anxiety and depressive symptoms, along with significantly elevated levels of stress hormones compared to healthy controls [25–27]. Further research has revealed that patients who do not respond to treatment tend to report more severe depression, anxiety, and stress responses, which can further exacerbate treatment outcomes, thereby creating a vicious cycle [28,29]. Therefore, we reasonably infer that in patients with presence of

SNHL in one ear, the occurrence of ISSNHL in the other ear may significantly increase anxiety and depression, which in turn may negatively influence the prognosis of ISSNHL. Consequently, during treatment, in addition to focusing on hearing recovery, attention should also be directed toward the assessment and intervention of the patients' mental health to improve overall prognosis.

To further analyze the impact of contralateral SNHL prior to the onset of ISSNHL on prognosis, we performed a subgroup analysis and divided the patients into two groups: the contralateral normal hearing group and the contralateral SNHL group. The results showed that the complete recovery rate and partial recovery rate in the contralateral normal hearing group were significantly higher than those in the contralateral SNHL group (p < 0.001, p = 0.015), while the no recovery rate was significantly lower in the contralateral normal hearing group. This further suggests that the presence of contralateral SNHL is an influencing factor for a poorer prognosis.

Based on the above analysis, we speculate that for ISSNHL patients,the presence of contralateral SNHL before the onset of ISSNHL may serve as an indicator of poor prognosis. This factor could thus function as an important predictive marker for prognosis. Our findings demonstrate that contralateral SNHL is an independent risk factor for poor prognosis in ISSNHL patients. This discovery offers clinicians a new perspective, underscoring the importance of considering the

hearing status of the contralateral ear when assessing patient prognosis. It also highlights the need to develop individualized treatment plans based on other clinical characteristics to optimize treatment outcomes and improve patients' quality of life.

However, this study has certain limitations. Although all patients underwent standardized examinations and treatments, the retrospective design may introduce selection bias. Additionally, the study did not investigate in detail the potential causes of the contralateral SNHL, such as noise exposure or ototoxic medications, which may have influenced the results. In this study, due to the reluctance of complete recovered patients to undergo further hearing evaluations, we did not conduct a formal hearing follow-up at 1 month. This limitation means that the long-term hearing changes in fully recovered patients were not included in the analysis. Future studies will adopt a prospective design with a larger sample size and systematically assess possible etiologies of contralateral hearing loss, in order to further validate and extend the findings of this study.

## Conclusion

This study specifically focused on the impact of contralateral SNHL on treatment outcomes, with the results clearly demonstrating a significant positive correlation between contralateral hearing loss and the risk of treatment failure in ISSNHL. This finding suggests that contralateral SNHL can serve as a crucial predictive marker for assessing the prognosis of ISSNHL. Since maximizing hearing recovery remains the ultimate goal of treatment, our research emphasizes the importance of closely monitoring the hearing status of the contralateral ear in the clinical evaluation and management of ISSNHL patients.

Additionally, we observed that contralateral SNHL interacts multiplicatively with factors such as age over 60 years, hypertension, and a disease duration exceeding 4 days, which may further increase the risk of treatment failure. Therefore, in clinical practice, incorporating individualized factors-such as disease duration, patient age, and hypertension-into the development of targeted treatment plans can help optimize treatment outcomes. In summary, through multidimensional assessment and personalized management, it is possible to offer ISSNHL patients more effective treatment strategies, ultimately improving their prognosis and quality of life.

## Author contributions

**Conceptualization:** Sanlin Xie, Zhifeng Chen.

**Data curation:** Sanlin Xie, Zhifeng Chen.

**Formal analysis:** Zhifeng Chen, Lipeng Huang.

**Funding acquisition:** Chang Lin.

**Methodology:** Lipeng Huang.

**Supervision:** Yongjun Hong.

**Writing – original draft:** Sanlin Xie.

**Writing – review & editing:** Sanlin Xie, Yongjun Hong, Chang Lin.

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
