## [Decision Letter · Decision Letter 0]

30 Dec 2025

PONE-D-25-46934Impact of Contralateral Sensorineural Hearing Loss on Prognosis in Idiopathic Sudden Sensorineural Hearing LossPLOS One

Dear Dr. Xie,

Thank you for submitting your manuscript to PLOS ONE. After careful consideration, we feel that it has merit but does not fully meet PLOS ONE’s publication criteria as it currently stands. Therefore, we invite you to submit a revised version of the manuscript that addresses the points raised during the review process.

**ACADEMIC EDITOR:**Please reply to Reviewer 2 comments, revise your manuscript and re-submit

We look forward to receiving your revised manuscript.

Kind regards,

Gauri Mankekar, MD,PhD,FACS

Academic Editor

PLOS One

Journal Requirements:

Reviewers' comments:

Reviewer's Responses to Questions

**Comments to the Author**

1. Is the manuscript technically sound, and do the data support the conclusions?

Reviewer #1: Yes

Reviewer #2: Partly

2. Has the statistical analysis been performed appropriately and rigorously?

Reviewer #1: Yes

Reviewer #2: Yes

3. Have the authors made all data underlying the findings in their manuscript fully available?

Reviewer #1: Yes

Reviewer #2: Yes

4. Is the manuscript presented in an intelligible fashion and written in standard English?

Reviewer #1: Yes

Reviewer #2: Yes

5. Review Comments to the Author

Reviewer #1: Well written and analyzed. The authors make clear their limitations of not investigating the cause of hearing loss. That in many ways can influence treatment. That would be a very influential factor. Since this is a retrospective analysis, perhaps this could influence future prospective study designs

Reviewer #2: Issues and Questions to be Addressed

Throughout the Manuscript:

- Instead of using “effective/ineffective”, consider using something like “recovery/no-recovery”. Effective/ineffective could imply that there a prospective clinical trial, designed with a specific treatment protocol that includes a treatment or placebo control, was studied.

- Instead of stating “…the contralateral was…”, always state “…the contralateral ear ….”. Contralateral is usually used as an adjective, not a noun.

Introduction:

- Add references for prior research that considered the contralateral ear. Cite these at end of next-to-last paragraph in Introduction.

- Also, add references for last paragraph in Introduction, including after the statement, “Building upon existing evidence…”.

Statistical Methods and Variables:

- Consider magnitude of sudden hearing loss as a co-variable.

- Consider the time of treatment onset relative to the onset of the sudden hearing loss as a co-variate.

- Consider the magnitude of recovery in reference to the hearing threshold in the affected ear prior to the sudden hearing loss. See 2nd paragraph in Methods, Hearing Assessment.

- Consider ”diabetic peripheral neuropathy” as a co-morbidity.

- The use of ototoxic medications prior to onset of sudden hearing loss (within some time period) should be an exclusion criterion. Alternatively, use it as a variable in the statistical analyses.

Methods:

- Inclusion criteria: elaborate on “no prior treatment”.

- Change from “anonymized” to “de-identified”.

- Hearing assessment: Reconcile the dB used for mild and profound hearing loss with the audiogram patterns (e.g., >= 25 dB at 0.25 and 0.5 Hz for hearing loss threshold, and > 81 dB at all frequencies for total deafness).

- For complete recovery, why is “10 dB of the contralateral ear” used instead of “10 dB of baseline (pre-onset of sudden loss) of the affected ear itself”?

Results:

- For those with complete recovery (based on PTA at discharge), what was the PTA at 1 month post-discharge (as reported for those without complete recovery)? Any significant change at 1 month (compared to end of discharge)?

Discussion:

- Is there an example of a possible “individualized patient assessment protocols”, as proposed in the Introduction, in addition to “assessment and intervention of the patient’s mental health”.

6. PLOS authors have the option to publish the peer review history of their article (what does this mean?). If published, this will include your full peer review and any attached files.

Reviewer #1: **Yes:**Christopher de Souza

Reviewer #2: **Yes:**John H Anderson

---

## [Author Response · Author response to Decision Letter 1]

17 Jan 2026

Response to Reviewer #1:

Dear Reviewer,

Thank you for your positive and thoughtful feedback. We appreciate your recognition of our manuscript and the clear acknowledgment of its strengths.

Regarding your comment on the limitations of not investigating the cause of hearing loss:

We completely agree with your point that understanding the cause of hearing loss is an important factor that could influence treatment outcomes. In our study, we aimed to focus on the effects of treatment based on the data available, but we acknowledge that the cause of hearing loss may indeed have significant implications for how patients respond to treatment. We have now emphasized this limitation in the discussion section of the manuscript and highlighted that future prospective studies should consider the cause of hearing loss as an important variable when evaluating treatment options. This could be a valuable area for future research and could help design more tailored treatments based on the underlying cause.

Once again, we appreciate your constructive comment, and we hope that our response clarifies this aspect of our study. We believe that future research incorporating these factors will enhance the understanding and treatment of hearing loss.

Response to Reviewer #2

1.Instead of using “effective/ineffective”, consider using something like “recovery/no-recovery”. Effective/ineffective could imply that there a prospective clinical trial, designed with a specific treatment protocol that includes a treatment or placebo control, was studied.

Reply: Thanks for your suggestion. We have made revisions based on your suggestions.

2.Instead of stating “…the contralateral was…”, always state “…the contralateral ear ….”. Contralateral is usually used as an adjective, not a noun.

Reply: Thanks for your suggestion. We have made revisions based on your suggestions.

3.Add references for prior research that considered the contralateral ear. Cite these at end of next-to-last paragraph in Introduction.

Reply: Thanks for your suggestion. We have added the references (on line 70).

[3] Koo M, Hwang JH. Risk of sudden sensorineural hearing loss in patients with common preexisting sensorineural hearing impairment: a population-based study in Taiwan. PLoS One. 2015 Mar 27;10(3):e0121190. doi: 10.1371/journal.pone.0121190.

[4] Liu Y, Wu W, Li S, Zhang Q, He J, Duan M, et al. Clinical characteristics and prognosis of sudden sensorineural hearing loss in single-sided deafness patients. Front Neurol. 2023 Sep 27;14:1230340. doi: 10.3389/fneur.2023.1230340.

4. References for the last paragraph in Introduction

Reply: Thanks for your suggestion. We have added the references (on line 71).

[3] Koo M, Hwang JH. Risk of sudden sensorineural hearing loss in patients with common preexisting sensorineural hearing impairment: a population-based study in Taiwan. PLoS One. 2015 Mar 27;10(3):e0121190. doi: 10.1371/journal.pone.0121190.

[4] Liu Y, Wu W, Li S, Zhang Q, He J, Duan M, et al. Clinical characteristics and prognosis of sudden sensorineural hearing loss in single-sided deafness patients. Front Neurol. 2023 Sep 27;14:1230340. doi: 10.3389/fneur.2023.1230340.

5.Consider magnitude of sudden hearing loss as a co-variable

Reply: Thanks for your valuable suggestion regarding our study.

In the manuscript, We conducted a univariate analysis on the magnitude of sudden hearing loss and found no statistically significant difference in this factor. In the manuscript,we use 'pre-treatment PTA of the affected ear' to represent the magnitude of sudden hearing loss.Although many previous studies have suggested that the initial hearing threshold and degree of hearing loss are an important prognostic factor for ISSNHL, our study did not find a statistically significant relationship between this factor and prognosis. The possible reason for this is that our study is a retrospective analysis, and most of the patients who received treatment in our study had more severe hearing loss, while those with mild hearing loss typically received treatment in the outpatient department. Additionally, the focus of this study was to explore the impact of contralateral sensorineural hearing loss on prognosis in ISSNHL, and a detailed analysis of this aspect was conducted. Future research will continue to explore this direction and focus on prospective multi-center studies. (on line 288-294).

6. Consider the time of treatment onset relative to the onset of sudden hearing loss

Reply: Thanks for your suggestion. Regarding your recommendation to consider the time of treatment onset relative to the onset of sudden hearing loss as a co-variable, we have actually already accounted for this factor in the manuscript by using the definition of “duration of illness”, which refers to the interval from symptom onset to initiation of treatment. We believe this concept is essentially the same as the time of treatment onset relative to the onset of sudden hearing loss you mentioned.

In our analysis, we have included duration of illness as a co-variable and assessed its potential impact on treatment outcomes.

7.Consider the magnitude of recovery in reference to the affected ear's hearing threshold

Reply:Thank you for your valuable suggestion. In our study, we classified patients into two groups based on the hearing status of the contralateral ear: contralateralear normal hearing group and contralateral ear SNHL group. We found no statistically significant difference in the affected ear's hearing threshold (Pre-treatment PTA (affected ear)) between the two groups. However, there was a statistically significant difference between the two groups in terms of hearing gain and the magnitude of recovery.

This finding suggests that the magnitude of recovery is not significantly related to the affected ear's hearing threshold, but is closely associated with the hearing status of the contralateral ear. This further emphasizes the importance of considering factors such as the contralateral ear's hearing status when evaluating hearing recovery in patients with ISSNHL.(Table 5)

8.Diabetic peripheral neuropathy as a co-morbidity.

Reply: Thank you for your valuable suggestion. We have indeed considered diabetic peripheral neuropathy as a comorbidity in our study. Diabetic sensorineural hearing loss is one of the common symptoms of diabetic peripheral neuropathy, typically presenting as bilateral progressive sensorineural hearing loss. This could also be a potential factor contributing to the contralateral ear sensorineural hearing loss. However, due to data limitations and the nature of our retrospective study, we were unable to categorize the causes of the contralateral ear sensorineural hearing loss in detail, which is a limitation of our study. We have discussed this point in the limitations section. In future prospective studies, we will focus on this factor and further explore its impact on the prognosis of ISSNHL.Thank you again for your valuable suggestion, and we will address this issue in our future research.

9.Exclusion criterion for ototoxic medications

Reply: Thanks for your suggestion.According to your suggestions, we have made appropriate changes to the article.(on line 91-92)

10. Elaboration on 'no prior treatment' in inclusion criteria

Reply: Thanks for your suggestion.According to your suggestions, we have made appropriate changes to the article.(on line 88-89)

Patients referred to those receiving their first treatment at our hospital after symptom onset, with no prior interventions administered elsewhere

11. Change from 'anonymized' to 'de-identified'

Reply: Thanks for your suggestion. We have made revisions based on your suggestions.

12.Reconcile the dB values for mild and profound hearing loss with audiogram patterns

Reply: Thanks for your suggestion. We have made revisions based on your suggestions.(on line 103-115)

PTA was calculated as the average of the air-conduction thresholds at 0.5, 1, 2, and 4 kHz for the affected ear. The severity of hearing loss was classified as follows: mild (26 ≤ PTA < 41 dB), moderate (41 ≤ PTA < 61 dB), severe (61 ≤ PTA < 81 dB), and profound (PTA ≥ 81 dB). audiometric curves were categorized into four types based on the hearing threshold patterns[5]:Low-frequency type: Hearing loss primarily occurs at frequencies ≤ 1 kHz, with thresholds at 0.25 kHz and 0.5 kHz showing a hearing loss of ≥ 20 dB;High-frequency type: Hearing loss primarily occurs at frequencies ≥ 2 kHz, with thresholds at 4 kHz and 8 kHz showing a hearing loss of ≥ 20 dB; Flat type: Relatively uniform hearing loss is observed across all frequencies from 0.25 to 8 kHz (at 0.25, 0.5, 1, 2, 4, and 8 kHz), with a mean PTA ≤ 80 dB.

Total deafness type: Profound hearing loss is observed across all tested frequencies (0.25–8 kHz), with a mean PTA ≥ 81 dB.

13.Change from '10 dB of the contralateral ear' to '10 dB of baseline in the affected ear'

Reply: Thanks for your suggestion. We have modified the definition of complete recovery to reflect '10 dB of baseline (pre-onset) of the affected ear' instead of '10 dB of the contralateral ear.'

14.For those with complete recovery (based on PTA at discharge), what was the PTA at 1 month post-discharge (as reported for those without complete recovery)? Any significant change at 1 month (compared to end of discharge)?

Reply: Thanks for your suggestion. We have discussed this point in the limitations section.(on line 333-339)

In this study, due to the reluctance of complete recovered patients to undergo further hearing evaluations, we did not conduct a formal hearing follow-up at 1 month. This limitation means that the long-term hearing changes in fully recovered patients were not included in the analysis. Future studies will adopt a prospective design with a larger sample size and systematically assess possible etiologies of contralateral hearing loss, in order to further validate and extend the findings of this study.

15. Is there an example of a possible “individualized patient assessment protocols”, as proposed in the Introduction, in addition to “assessment and intervention of the patient’s mental health”.

Reply: Thanks for your suggestion.We have a case like this:personalized assessment for a 70-year-old male patient with diabetes and PTA > 80 dB.Considering the patient's advanced age and diabetes, special attention should be given to the impact of medications on blood glucose levels, and the use of systemic corticosteroids should be minimized.Prioritize low-dose corticosteroid treatment, monitor blood glucose levels, and adjust medications as needed. Severity of hearing loss: with a PTA > 80 dB, the patient has severe hearing loss and requires a more aggressive treatment strategy.Use intratympanic corticosteroid injections in combination with oral corticosteroids to increase drug concentration in the affected ear and reduce systemic side effects.

---

## [Editor Report · Decision Letter 1]

2 Mar 2026

Impact of Contralateral Sensorineural Hearing Loss on Prognosis in Idiopathic Sudden Sensorineural Hearing Loss

PONE-D-25-46934R1

Dear Dr. Sanlin Xie,

Thank you for responding to the reviewer's comments and submitting the revised manuscript. We’re pleased to inform you that your manuscript has been judged scientifically suitable for publication and will be formally accepted for publication once it meets all outstanding technical requirements.

Kind regards,

Gauri Mankekar, MD,PhD,FACS

Academic Editor

PLOS One

---

## [Editor Report · Acceptance letter]

PONE-D-25-46934R1

PLOS One

Dear Dr. Hong,

I'm pleased to inform you that your manuscript has been deemed suitable for publication in PLOS One. Congratulations! Your manuscript is now being handed over to our production team.

Kind regards,

on behalf of

Dr. Gauri Mankekar

Academic Editor

PLOS One